# Association of family structure with gain and loss of household headship among older adults in India: Analysis of panel data

**Shobhit Srivastava**[1]ᵒ, **Muhammad Thalil**[2]ᵒ, **Rashmi Rashmi**[3]ᵒ, **Ronak Paul**[3]ᵒ*

**1** Department of Mathematical Demography and Statistics, International Institute for Population Sciences, Mumbai, Maharashtra, India, **2** Department of Population Policies and Programmes, International Institute for Population Sciences, Mumbai, Maharashtra, India, **3** Department of Public Health and Mortality Studies, International Institute for Population Sciences, Mumbai, Maharashtra, India

ᵒ These authors contributed equally to this work.
\* greenophenn@gmail.com

## Abstract

### Background

Despite huge changes in demographic behaviors, the family continues to be the major source of psychosocial support for older adults in India. The loss of household headship can be a cause of disregard for the aged and is associated in a very fundamental way with other status losses. Our study used the two rounds of the India Human Development Survey to understand the association of family structure on the gain or loss status of household headship among 10,527 older adults.

### Method

Bivariate analysis was done using the chi-square test for association. Equivalently, the multivariate analysis involved estimating multivariable logistic regression models. Multicollinearity did not affect the estimates from the regression models. For examining headship transition, we performed two complete sets of analysis, by taking gain in headship and loss in headship as the outcome variable respectively.

### Results

Across two rounds, a major shift in family structure was noticed as 6.8% of households moved from extended to a single generation. Results indicate that family structure was significantly associated with gaining and losing headship among older adults. Headship loss was more common among nuclear [OR: 2.16; CI: 1.28, 3.65] and extended [OR: 2.76; CI: 1.64, 4.66] family structures. Moreover, gaining headship was found to be significantly associated with married, educated, and working older adults.

### Conclusion

Since living in single generation household may preferably be encouraged among older adults than their living in a complex household without headship and value they deserve, the

**Data Availability Statement:** This study used a publicly available secondary dataset with no information that could lead to the identification of the respondents. The IHDS datasets used in our

study can be downloaded from the Inter-University Consortium for Political and Social Research (ICPSR) data repository [https://www.icpsr.umich.edu/web/DSDR/series/507].

**Funding:** The author(s) received no specific funding for this work.

**Competing interests:** The authors have declared that no competing interests exist.

public intervention may support the independent living within the older population through housing policies that create additional choices presented to older adults making residential decisions.

## Introduction

Demographic transition coupled with rapid social changes poses a serious threat to the aging population in India. Changing family demography has affected the well-being of older adults by depriving them of familial support and changes in values and norms in society [1, 2]. On the other hand, living as a household head signifies a greater degree of independence than living in a household under someone else's authority [3]. However, the shift in the status of the aged from dominant to subordinate position occurs quite prominently with the advancing age [4]. Understanding the potential factors that could have accounted for such changes in household headship status of older adults may help improve policy and services that would support and uphold their role and authority in the family system.

Household headship is seen as a reflection of both income-earning as well as decision-making status in the household [5–7]. As an index of family status, headship signifies the importance of the family member concerning their power to control and allocate the family's economic and social resources, a greater degree of economic and symbolic power [8]. On the other hand, the loss of headship can be a cause of disregard for the aged and is associated in a very fundamental way with other status losses such as retirement and widowhood that mark the transition into the final life stage [9]. Interestingly, the headship status in Asian countries is unstable. For example, independent living is preferred as one phase of the life course while having adequate economic resources [10], and that would give way to the traditional form of co-residential living son's or daughter's headship when the older parent becomes frail and chronically ill [11].

Earlier studies revealed that heading an independent household is a function of age, marital status, parental status, and individual income [12]. Economic resources including pension receipts are also seen as helping the older parents to maintain their locus of power in the household, which could be eroded by becoming a burden to other members in an extended family system [13]. Another study in Thailand found that the extensive rural to urban migration of adult children have not led to the widespread desertion of left-behind older parents unless frailty and poor health required daily personal assistance [14]. Nonetheless, as couples grow too old and infirm to maintain separate households, they either enter the households of their children or a nursing home and other institutions [15]. In the case of unmarried older adults, they are left to choose among living arrangements that are much complex such as with extended kin, nonrelatives or they may live alone [8].

### Family structure and household headship

In Asian countries including India, strong family values are maintained; with many people living in extended family households either together or close by for their psychological, social, and physical needs [16]. Once settled, family headship continues over the family life course unless critical events, such as divorce or migration, disrupt the family structure. Despite the strong stability of family headship, age-related events such as the decline in health and death of spouse limit older persons' ability to maintain their independence and require adjustments in their living arrangements that will affect their headship roles [17].

The family in India is considered as a group of people that includes both the living generations immediately above and below the household head and their predecessors and those who are yet to be born [18]. The majority of older people live with their immediate family members and the family continues to be the main provider of care for the aged [19]. Notably, the Indian joint family has always been regarded as an ideal group of people that is bounded by the loyalties towards the members living together in one household under the authority of an older person [20]. Furthermore, the heads of most of the families are older people and they are cared for and respected especially, those who were medically healthy and performed their daily activities independently [21]. When parents and unmarried siblings are working, married sons feel that there is no reason for them to delay entering the life stage defining masculine adulthood, that of the head of household. They do so by setting up their own nuclear family frequently by requiring parents to subdivide the family home [22, 23] hence, the older parents lose their household headship.

In the Indian context, despite huge changes in demographic behaviors, the family continues to be the major source of psychosocial support for older adults. Hence, there arises the need to understand how transitions in family structures affect the experiences of headship changes among older adults. More than a cross-sectional view, there is a necessity to examine such association over time. Our study had tried to explore this gap in the literature and used the panel dataset to examine the effect of family structure in elevating the chances of gain or loss of household headship among older adults within a period of seven years.

## Data and methods

### Data source

The current study used the India Human Development Survey (IHDS) wave-I and wave-II. IHDS wave-I is a nationally representative and multi-topic survey of 41,554 households conducted during 2004–05 across all the states and union territories of India excluding the Andaman & Nicobar Islands and Lakshadweep [24]. IHDS wave-II conducted during 2011–12 is also a nationally representative and multi-topic survey of 42,152 households with geographical coverage similar to wave-I [25]. Both waves of IHDS adopted a multistage stratified random sampling approach. IHDS wave-II re-interviewed 83% of the households from wave-I. Further details regarding IHDS sampling and data collection procedures can be found elsewhere [26, 27].

During IHDS wave-I, there were 17,906 individuals, aged 60 years and above, whom we refer to as older adults in this study. Among them, 4,736 older adults were not alive and 2,643 older adults were lost to follow-up during wave-II. Thus, our current study is based on panel data of 10,527 older adults. Additionally, there were no records with missing information for all the variables used in our study.

### Ethics statement

Our study utilized publicly available secondary datasets with no information that could lead to the identification of the respondents. Therefore, prior ethical approval for using these datasets was not necessary. Further, these datasets were collected and owned by a third-party and the authors did not have any special privileges for accessing these datasets.

### Outcome variables

We used two binary outcome variables in this study. The first indicator shows that whether older adults who were not household heads during wave-I gained headship during wave-II.

The second indicator measures whether older adults who were household heads during wave-I had lost headship during wave-II. Both these outcome indicators were obtained from every individual's respective headship status during both waves of IHDS. In the gained headship variable, older adults who were "not head" in wave-I and became "head" in wave-II were coded as "yes", and if they were "not head" in both waves then they were coded as "no". Similarly, in the lost headship variable older adults who were "head" during wave-I but became "not head" during wave-II were coded as "yes", and those who were "head" in both waves were coded as "no".

## Explanatory variables

Household family structure during wave-I is the main explanatory variable of this study. The Household family structure variable was obtained from the information given on the relationship of each household member with the head of the household. Based on this information the family structure was categorized into three family types–single generation, nuclear, and joint/extended family. The single generation includes a married/cohabiting couple or a single-person household. The nuclear family includes married/cohabiting partners along with their dependent and unmarried children. The joint family includes a parent and/or partner along with their children and grandchildren. The extended family is similar to a joint family structure with the exception that it also includes "extended members", that is, people who are not directly related to the household head by blood. Extended relatives include brother-in-law, sister-in-law, daughter-in-law, son-in-law, stepfather, stepmother, parent-in-law, and servants.

## Control variables

Existing studies show several factors other than family structure, which affects the transition of household headship among older adults. We controlled for the confounding effects of the majority of these variables in our study conditional to their availability in IHDS. The control variables related to the individual older adults include–marital status (not currently married, currently married), level of education (no formal schooling, less than 5 years of schooling, 6–10 years of schooling, more than 10 years of schooling), age group (60–69 years, 70–79 years, 80+ years), working status (not working, working), whether received old-age pension (no, yes), whether suffering from chronic diseases (no, yes). We also controlled for relevant demographic, socio-economic and geographic characteristics that are–number of married adults in household (0, 1, 2, 3, 4, 5 and more), number of unmarried adults in household (0, 1, 2, 3, 4, 5 and more), number of children in household (0, 1, 2, 3, 4, 5 and more), gender of the household head (male, female), household wealth quintile (poorest, poor, medium, rich, richest), household below poverty line (BPL) status (not poor, poor), the caste of the household (scheduled tribes (ST), scheduled castes (SC), other backward classes (OBC), others), the religion of the household (Hindu, Muslim, others), place of residence (rural, urban), country regions–(northern, north-eastern, central, eastern, western, southern). All the above characteristics were measured for the older adults during wave-I.

During wave-I IHDS collected information on the marital status of each individual from a household. The marital status was originally categorized into six categories–"spouse absent", "married", "single", "widowed", "separated/divorced" and "no gauna". Judging by the percentage of individuals in each category, we have recoded the original variable into a binary marital status variable. Individuals who were "single", "widowed", "separated/divorced", "spouse absent" and "no gauna" were recoded into the "Not currently married" category; otherwise, they were included in the "Currently married" category.

The variable of whether an older adult suffers from any other chronic diseases during wave-I were obtained from the information on whether each individual suffered from–cataract,

tuberculosis, high blood pressure, cardiovascular diseases, diabetes, leprosy, cancer, asthma, polio, paralysis, epilepsy, mental illness, sexually transmitted diseases (STD) and any other chronic disease. If an older adult suffered from at least one of the above chronic diseases, then they were coded as "yes" and otherwise they were coded as "no".

The household wealth quintile for wave-I was calculated using principal component analysis [28]. We generated wealth scores for each household using the available information on household asset ownership, livestock ownership, building material used in household, household water source, household sanitation facility and the number of rooms. Based on the wealth score we categorized the households into five categories (poorest, poor, medium, rich, richest) such that the households with the lowest 20 percentile score belonged to the "poorest" category, households with the next low 20 percentile score belonged to the "poor" category and so forth.

The country regions during wave-I were formed by dividing the erstwhile 33 states and union territories of India into six regions. The northern region includes Chandigarh, Delhi, Haryana, Himachal Pradesh, erstwhile Jammu & Kashmir, Punjab, Uttaranchal and Rajasthan. The north-eastern region includes Assam, Arunachal Pradesh, Manipur, Meghalaya, Mizoram, Nagaland, Tripura and Sikkim. The central region consists of Madhya Pradesh and Chhattisgarh. The eastern region includes Bihar, Jharkhand, Odisha and West Bengal. The western region comprises Dadra & Nagar Haveli, Daman & Diu, Goa, Gujarat and Maharashtra. The southern region comprises erstwhile Andhra Pradesh, Karnataka, Kerala, Tamil Nadu and Pondicherry.

## Statistical methods

We performed bivariate and multivariate analysis to achieve the study objectives. Owing to the binary nature of the outcome variable, bivariate analysis was done using the chi-square test for association. Equivalently, multivariate analysis was performed by estimating multiple variable logistic regression models. Bivariate and multivariate analysis was performed in two sets by taking "gained headship" and "lost headship" during wave-II as the outcome variable respectively. Odds ratios in the multivariable models show the association between the outcome variables of transition in headship and the independent variables. The odds ratio measures the odds (chance) of losing headship (or gaining headship) relative to having an unchanged headship status among the older adults belonging to a particular category of an explanatory variable given the effect of all the other explanatory variables remain constant [29]. The odds ratio can take any value above zero, with a value between 0 and 1 denoting a negative association, and a value more than 1 denoting a positive association.

Additionally, we checked for multicollinearity in both the regression models and found the mean value of the variance inflation factor (VIF) to be less than 1.9. Thus, our estimated regression models do not suffer from multicollinearity. The use of panel data requires the application of panel weights. Unfortunately, the results given in this study are unweighted, as IHDS does not provide separate panel weights for analysis. All the statistical estimations were done using the STATA 13 software [30].

## Results

### Descriptive findings

Fig 1 depicts that 8.3%, 3.9% and 67.6% of older adults residing in a single, nuclear and joint family structure in wave-I respectively, had experienced no transition in their family structure from wave-I to wave-II. Further, 0.1% and 2.5% of older adults had moved from a single generation to a nuclear and extended family structure respectively across the two rounds. Nearly,

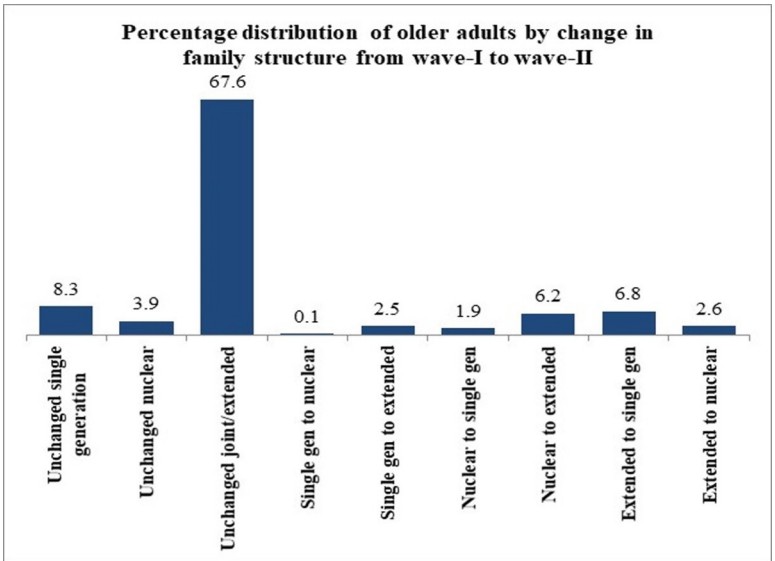

**Fig 1. Distribution of older adults by change in family structure from wave-I to wave-II.**

6.2% of older adults had experienced a change in family structure from nuclear to extended, while 6.8% of older adults had moved from extended to single generation family structure.

As shown in Fig 2, the percentage of older adults who were residing in a single generation household in wave-I increased to nearly 6% in wave-II. In contrast, older adults residing in the nuclear family structure had decreased from 12.1% in wave-I to 6.7% in wave-II. About 1% decrement was also seen among the older adults residing in joint or extended family structure from wave-I to wave-II.

Table 1 shows the characteristics of 17,904 and 10,527 older adults respectively in the cross-sectional and panel datasets during wave-I. In the panel dataset, 77% of older adults resided in

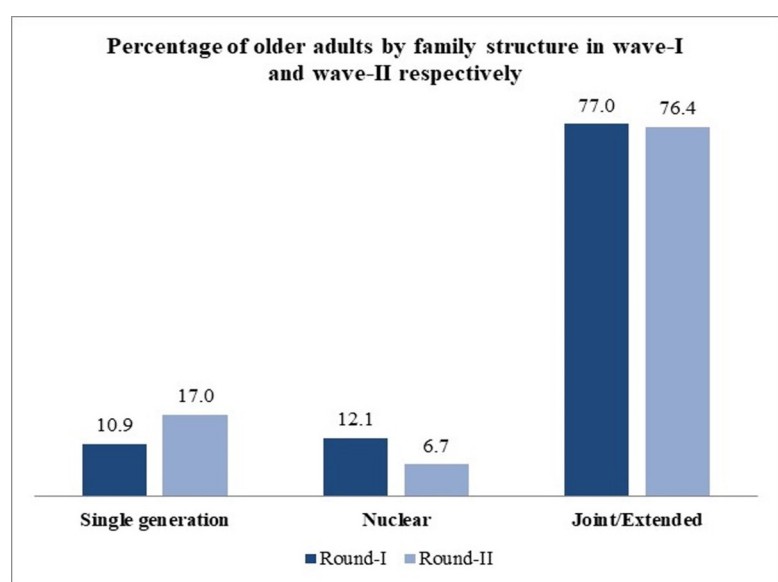

**Fig 2. Distribution of older adults by their family structure during waves -I and–II.**

**Table 1. Distribution of older adults by socio-economic and demographic characteristics across cross-sectional and panel datasets during wave-I.**

| Characteristics in wave-I | IHDS wave-I | | | | Absolute difference |
|---|---|---|---|---|---|
| | Cross-sectional dataset | | Panel dataset | | |
| | N | % | N | % | % |
| **Family structure** | | | | | |
| Single generation | 2,045 | 11.4 | 1,150 | 10.9 | 0.5 |
| Nuclear | 2,016 | 11.3 | 1,268 | 12.0 | 0.7 |
| Joint/Extended | 13,843 | 77.3 | 8,109 | 77.0 | 0.3 |
| **Marital status of individual** | | | | | |
| Not currently married | 6,650 | 37.1 | 3,441 | 32.7 | 4.4 |
| Currently married | 11,254 | 62.9 | 7,086 | 67.3 | 4.4 |
| **Level of education of individual** | | | | | |
| No formal schooling | 10,672 | 59.6 | 6,262 | 59.5 | 0.1 |
| Less than 5 years of schooling | 3,128 | 17.5 | 1,867 | 17.7 | 0.2 |
| 6–10 years of schooling | 2,736 | 15.3 | 1,622 | 15.4 | 0.1 |
| More than 10 years of schooling | 1,368 | 7.6 | 776 | 7.4 | 0.2 |
| **Age group (in years)** | | | | | |
| 60–69 | 10,917 | 61.0 | 7,343 | 69.8 | 8.8 |
| 70–79 | 5,200 | 29.0 | 2,663 | 25.3 | 3.7 |
| 80+ | 1,787 | 10.0 | 521 | 4.9 | 5.1 |
| **Working status of individual** | | | | | |
| Not working | 15,497 | 86.6 | 8,934 | 84.9 | 1.7 |
| Working | 2,407 | 13.4 | 1,593 | 15.1 | 1.7 |
| **Individual received old-age pension** | | | | | |
| No | 16,483 | 92.1 | 9,825 | 93.3 | 1.2 |
| Yes | 1,421 | 7.9 | 702 | 6.7 | 1.2 |
| **Individual suffering from chronic diseases** | | | | | |
| No | 14,312 | 79.9 | 8,683 | 82.5 | 2.6 |
| Yes | 3,592 | 20.1 | 1,844 | 17.5 | 2.6 |
| **Gender of household head** | | | | | |
| Male | 16,287 | 91.0 | 9,573 | 90.9 | 0.1 |
| Female | 1,617 | 9.0 | 954 | 9.1 | 0.1 |
| **Household wealth quintile** | | | | | |
| Poorest | 2,699 | 15.1 | 1,603 | 15.2 | 0.1 |
| Poor | 2,961 | 16.5 | 1,777 | 16.9 | 0.4 |
| Medium | 3,491 | 19.5 | 2,109 | 20.0 | 0.5 |
| Rich | 3,852 | 21.5 | 2,241 | 21.3 | 0.2 |
| Richest | 4,901 | 27.4 | 2,797 | 26.6 | 0.8 |
| **Household BPL status** | | | | | |
| Not poor | 14,312 | 79.9 | 8,445 | 80.2 | 0.3 |
| Poor | 3,592 | 20.1 | 2,082 | 19.8 | 0.3 |
| **Caste of household** | | | | | |
| Scheduled Tribes | 1,126 | 6.3 | 607 | 5.8 | 0.5 |
| Scheduled Castes | 3,167 | 17.7 | 1,858 | 17.6 | 0.1 |
| Other Backward Classes | 7,191 | 40.2 | 4,334 | 41.2 | 1.0 |
| Others | 6,420 | 35.9 | 3,728 | 35.4 | 0.5 |
| **Religion of household** | | | | | |
| Hindu | 14,641 | 81.8 | 8,676 | 82.4 | 0.6 |
| Muslim | 1,758 | 9.8 | 994 | 9.4 | 0.4 |

*(Continued)*

**Table 1.** (Continued)

| Characteristics in wave-I | IHDS wave-I | | | | Absolute difference |
| --- | --- | --- | --- | --- | --- |
| | Cross-sectional dataset | | Panel dataset | | |
| | N | % | N | % | % |
| Others | 1,505 | 8.4 | 857 | 8.1 | 0.3 |
| **Place of residence** | | | | | |
| Rural | 12,647 | 70.6 | 7,763 | 73.7 | 3.1 |
| Urban | 5,257 | 29.4 | 2,764 | 26.3 | 3.1 |
| **Country regions** | | | | | |
| Northern | 5,771 | 32.2 | 3,488 | 33.1 | 0.9 |
| North Eastern | 562 | 3.1 | 290 | 2.8 | 0.3 |
| Central | 1,728 | 9.7 | 1,029 | 9.8 | 0.1 |
| Eastern | 2,637 | 14.7 | 1,599 | 15.2 | 0.5 |
| Western | 2,577 | 14.4 | 1,584 | 15.0 | 0.6 |
| Southern | 4,629 | 25.9 | 2,537 | 24.1 | 1.8 |
| **Overall** | **17,904** | **100** | **10,527** | **100** | **0** |

a joint or extended family during wave-I. Further, 67% of older adults were married, 60% had no formal education and 85% were unemployed during wave-I. Additionally, 70% of older adults belonged to the age group of 60–69 years and only 6.7% of older adults received an old-age pension. 91% of older adults come from a household with a male head, 20% belonged to a poor household. While 74% of older adults reside in rural India, a majority of the older adults belong to the northern (33%) and southern (24%) regions of India respectively.

Overall, we observed that the percentage distribution of older adults by the socio-economic and demographic characteristics was indeed similar in cross-sectional and panel datasets. Only percentage distribution by marital status, age group and place of residence differed by more than 3% points between the two datasets.

## Bivariate association

From Table 2, out of the panel of 10,527 older adults, 5,334 and 5,193 persons were "not head" and "head" of the household in wave-I respectively. Among 5,334 older adults who were not head in wave-I, 657 older adults have gained headship during wave-II. Further, among 5,193 older adults who were head in wave-I, 635 older adults lost their headship during wave-II. Among older adults who resided in a single generation family in wave-I, 25.5% had gained and 5.6% had lost their headship in wave-II. Almost 26% of older adults who reside in the nuclear family structure in wave-I had gained their headship during wave-II, in contrast to 7.8% of older adults who lost their headship. Further, only 10.5% of older adults from a joint or extended family structure in wave-I were able to gain their headship in wave-II. Gaining head-ship was common among older adults who were married (16.6%) and had 6–10 years of schooling education (17.1%) in wave-I. About 16% of households with no married adults shows gain in headship among older adults. Almost 14.1% older adults in the age group 60–69 years in wave-I had gained the headship in wave-II and 24.9% older adults who were 80 years and above in wave-I had lost their headship by wave-II. Around 8% of working older adults and 14.6% older adults who had a chronic disease in wave-I had lost their headship in wave-II. 27.2% of female older adults who were head in wave-I had lost their headship in wave-II. Approximately 15.5% of older adults from the north-eastern region in wave-I had gained their headship, while the loss in headship was mostly observed in southern regions (17.4%).

**Table 2. Bivariate analysis showing the association of socio-economic and demographic in wave-I with the gain and loss of headship in wave-II.**

| Characteristics in wave-I | Change in headship status during wave-II | | | | | | | |
|---|---|---|---|---|---|---|---|---|
| | Total | Gained headship | | Chi-square test for association | Total | Lost headship | | Chi-square test for association |
| | N | N | % | | N | N | % | |
| **Family structure** | | | | | | | | |
| Single generation | 388 | 99 | 25.5 | χ2 = 121.84; p-value = 0.000 | 762 | 43 | 5.6 | χ2 = 73.47; p-value = 0.000 |
| Nuclear | 258 | 67 | 26.0 | | 1,010 | 79 | 7.8 | |
| Joint/Extended | 4,688 | 491 | 10.5 | | 3,421 | 513 | 15.0 | |
| **Marital status of individual** | | | | | | | | |
| Not currently married | 2,248 | 145 | 6.5 | χ2 = 123.84; p-value = 0.000 | 1,193 | 298 | 25.0 | χ2 = 234.63; p-value = 0.000 |
| Currently married | 3,086 | 512 | 16.6 | | 4,000 | 337 | 8.4 | |
| **Number of married adults in household** | | | | | | | | |
| 0 | 158 | 25 | 15.8 | χ2 = 6.68; p-value = 0.245 | 456 | 77 | 16.9 | χ2 = 38.10; p-value = 0.000 |
| 1 | 41 | 6 | 14.6 | | 89 | 25 | 28.1 | |
| 2 | 2,423 | 303 | 12.5 | | 2,124 | 225 | 10.6 | |
| 3 | 181 | 22 | 12.2 | | 239 | 26 | 10.9 | |
| 4 | 1,802 | 199 | 11.0 | | 1,572 | 183 | 11.6 | |
| 5 and more | 729 | 102 | 14.0 | | 713 | 99 | 13.9 | |
| **Number of unmarried adults in household** | | | | | | | | |
| 0 | 485 | 111 | 22.9 | χ2 = 77.69; p-value = 0.000 | 721 | 35 | 4.9 | χ2 = 71.96; p-value = 0.000 |
| 1 | 560 | 86 | 15.4 | | 957 | 88 | 9.2 | |
| 2 | 968 | 134 | 13.8 | | 1,061 | 130 | 12.3 | |
| 3 | 1,155 | 124 | 10.7 | | 909 | 143 | 15.7 | |
| 4 | 860 | 79 | 9.2 | | 642 | 91 | 14.2 | |
| 5 and more | 1,306 | 123 | 9.4 | | 903 | 148 | 16.4 | |
| **Number of children in household** | | | | | | | | |
| 0 | 1,459 | 267 | 18.3 | χ2 = 68.00; p-value = 0.000 | 2,168 | 207 | 9.5 | χ2 = 46.22; p-value = 0.000 |
| 1 | 930 | 94 | 10.1 | | 908 | 97 | 10.7 | |
| 2 | 1,217 | 124 | 10.2 | | 875 | 146 | 16.7 | |
| 3 | 819 | 89 | 10.9 | | 576 | 86 | 14.9 | |
| 4 | 458 | 44 | 9.6 | | 314 | 37 | 11.8 | |
| 5 and more | 451 | 39 | 8.6 | | 352 | 62 | 17.6 | |
| **Level of education of individual** | | | | | | | | |
| No formal schooling | 3,904 | 444 | 11.4 | χ2 = 16.39; p-value = 0.001 | 2,358 | 365 | 15.5 | χ2 = 56.75; p-value = 0.000 |
| Less than 5 years of schooling | 705 | 103 | 14.6 | | 1,162 | 139 | 12.0 | |
| 6–10 years of schooling | 449 | 77 | 17.1 | | 1,173 | 103 | 8.8 | |
| More than 10 years of schooling | 276 | 33 | 12.0 | | 500 | 28 | 5.6 | |
| **Age group (in years)** | | | | | | | | |
| 60–69 | 3,603 | 507 | 14.1 | χ2 = 33.98; p-value = 0.000 | 3,740 | 348 | 9.3 | χ2 = 112.40; p-value = 0.000 |
| 70–79 | 1,419 | 131 | 9.2 | | 1,244 | 235 | 18.9 | |
| 80+ | 312 | 19 | 6.1 | | 209 | 52 | 24.9 | |
| **Working status of individual** | | | | | | | | |
| Not working | 4,971 | 583 | 11.7 | χ2 = 23.48; p-value = 0.000 | 3,963 | 537 | 13.6 | χ2 = 27.26; p-value = 0.000 |
| Working | 363 | 74 | 20.4 | | 1,230 | 98 | 8.0 | |
| **Individual received old age pension** | | | | | | | | |
| No | 4,945 | 606 | 12.3 | χ2 = 0.24; p-value = 0.621 | 4,880 | 576 | 11.8 | χ2 = 13.61; p-value = 0.000 |
| Yes | 389 | 51 | 13.1 | | 313 | 59 | 18.8 | |
| **Individual suffering from chronic diseases** | | | | | | | | |

*(Continued)*

**Table 2.** (Continued)

| Characteristics in wave-I | Change in headship status during wave-II | | | | | | |
|---|---|---|---|---|---|---|---|
| | Total | Gained headship | | Chi-square test for association | Total | Lost headship | | Chi-square test for association |
| | N | N | % | | N | N | % | |
| No | 4,426 | 556 | 12.6 | χ2 = 1.44; p-value = 0.229 | 4,257 | 498 | 11.7 | χ2 = 6.17; p-value = 0.013 |
| Yes | 908 | 101 | 11.1 | | 936 | 137 | 14.6 | |
| **Gender of household head** | | | | | | | | |
| Male | 5,175 | 631 | 12.2 | χ2 = 2.47; p-value = 0.116 | 4,398 | 419 | 9.5 | χ2 = 195.26; p-value = 0.000 |
| Female | 159 | 26 | 16.4 | | 795 | 216 | 27.2 | |
| **Household wealth quintile** | | | | | | | | |
| Poorest | 820 | 114 | 13.9 | χ2 = 4.26; p-value = 0.371 | 783 | 96 | 12.3 | χ2 = 1.18; p-value = 0.881 |
| Poor | 903 | 121 | 13.4 | | 874 | 115 | 13.2 | |
| Medium | 1,085 | 125 | 11.5 | | 1,024 | 123 | 12.0 | |
| Rich | 1,151 | 136 | 11.8 | | 1,090 | 135 | 12.4 | |
| Richest | 1,375 | 161 | 11.7 | | 1,422 | 166 | 11.7 | |
| **Household BPL status** | | | | | | | | |
| Not poor | 4,192 | 520 | 12.4 | χ2 = 0.14; p-value = 0.710 | 4,253 | 482 | 11.3 | χ2 = 17.53; p-value = 0.000 |
| Poor | 1,142 | 137 | 12.0 | | 940 | 153 | 16.3 | |
| **Caste of household** | | | | | | | | |
| Scheduled Tribes | 293 | 50 | 17.1 | χ2 = 10.41; p-value = 0.015 | 314 | 33 | 10.5 | χ2 = 11.14; p-value = 0.011 |
| Scheduled Castes | 939 | 128 | 13.6 | | 919 | 114 | 12.4 | |
| Other Backward Classes | 2,215 | 247 | 11.2 | | 2,119 | 294 | 13.9 | |
| Others | 1,887 | 232 | 12.3 | | 1,841 | 194 | 10.5 | |
| **Religion of household** | | | | | | | | |
| Hindu | 4,427 | 541 | 12.2 | χ2 = 0.32; p-value = 0.851 | 4,249 | 527 | 12.4 | χ2 = 1.08; p-value = 0.583 |
| Muslim | 489 | 61 | 12.5 | | 505 | 61 | 12.1 | |
| Others | 418 | 55 | 13.2 | | 439 | 47 | 10.7 | |
| **Place of residence** | | | | | | | | |
| Rural | 4,023 | 473 | 11.8 | χ2 = 4.75; p-value = 0.029 | 3,740 | 458 | 12.2 | χ2 = 0.01; p-value = 0.949 |
| Urban | 1,311 | 184 | 14.0 | | 1,453 | 177 | 12.2 | |
| **Country regions** | | | | | | | | |
| Northern | 1,786 | 228 | 12.8 | χ2 = 4.13; p-value = 0.531 | 1,702 | 176 | 10.3 | χ2 = 60.34; p-value = 0.000 |
| North Eastern | 116 | 18 | 15.5 | | 174 | 10 | 5.7 | |
| Central | 552 | 72 | 13.0 | | 477 | 68 | 14.3 | |
| Eastern | 764 | 80 | 10.5 | | 835 | 103 | 12.3 | |
| Western | 854 | 105 | 12.3 | | 730 | 56 | 7.7 | |
| Southern | 1,262 | 154 | 12.2 | | 1,275 | 222 | 17.4 | |
| **Overall** | **5,334** | **657** | **12.3** | | **5,193** | **635** | **12.2** | |

Note–(a) $\chi^2$: the value of the chi-square test statistic.

## Multivariate association

Estimates from two different logit models showing the change in the headship status of older adults in the household are shown in Table 3. Older adults who came from a joint or extended family in wave-I were less likely to gain headship in wave-II [OR: 0.66; CI: 0.32,1.38], in comparison to those who belonged to a single generation family in wave-I. Moreover, older adults coming from nuclear and joint/extended family structure in wave-I were 2.16 [CI: 1.28,3.65] and 2.76 [CI: 1.64,4.66] times higher likelihood to lose their headship respectively in wave-II.

**Table 3. Logistic regression estimates of socio-economic and demographic characteristics in wave-I with gain and loss in headship in wave-II.**

| Characteristics in wave-I | Change in headship status during wave-II | | | | | |
| --- | --- | --- | --- | --- | --- | --- |
| | Gained headship | | | Lost headship | | |
| | OR | 95% CI | P-value | OR | 95% CI | P-value |
| **Family structure** | | | | | | |
| Single generation | Ref | | | Ref | | |
| Nuclear | 0.73 | [0.35, 1.55] | 0.420 | 2.16 | [1.28, 3.65] | 0.004 |
| Joint/Extended | 0.66 | [0.32, 1.38] | 0.270 | 2.76 | [1.64, 4.66] | 0.000 |
| **Marital status of individual** | | | | | | |
| Not currently married | Ref | | | Ref | | |
| Currently married | 3.57 | [2.56, 4.97] | 0.000 | 0.35 | [0.25, 0.49] | 0.000 |
| **Number of married adults in household** | | | | | | |
| 0 | Ref | | | Ref | | |
| 1 | 0.87 | [0.31, 2.42] | 0.790 | 2.33 | [1.24, 4.39] | 0.009 |
| 2 | 0.51 | [0.30, 0.86] | 0.013 | 1.78 | [1.18, 2.68] | 0.006 |
| 3 | 0.43 | [0.20, 0.91] | 0.027 | 1.79 | [0.93, 3.43] | 0.082 |
| 4 | 0.31 | [0.16, 0.59] | 0.000 | 2.39 | [1.37, 4.16] | 0.002 |
| 5 and more | 0.50 | [0.25, 1.01] | 0.052 | 2.89 | [1.54, 5.40] | 0.001 |
| **Number of unmarried adults in household** | | | | | | |
| 0 | Ref | | | Ref | | |
| 1 | 1.27 | [0.64, 2.55] | 0.490 | 1.03 | [0.63, 1.69] | 0.890 |
| 2 | 1.55 | [0.78, 3.05] | 0.210 | 1.16 | [0.66, 2.03] | 0.610 |
| 3 | 1.50 | [0.74, 3.02] | 0.260 | 1.21 | [0.67, 2.18] | 0.520 |
| 4 | 1.32 | [0.63, 2.76] | 0.470 | 0.94 | [0.50, 1.77] | 0.860 |
| 5 and more | 1.69 | [0.79, 3.62] | 0.180 | 1.16 | [0.60, 2.25] | 0.650 |
| **Number of children in household** | | | | | | |
| 0 | Ref | | | Ref | | |
| 1 | 0.69 | [0.50, 0.95] | 0.025 | 0.71 | [0.51, 0.99] | 0.042 |
| 2 | 0.72 | [0.52, 1.00] | 0.050 | 1.08 | [0.76, 1.54] | 0.680 |
| 3 | 0.74 | [0.50, 1.10] | 0.140 | 0.95 | [0.62, 1.46] | 0.820 |
| 4 | 0.57 | [0.34, 0.95] | 0.030 | 0.70 | [0.40, 1.22] | 0.210 |
| 5 and more | 0.42 | [0.23, 0.74] | 0.003 | 1.11 | [0.62, 1.97] | 0.730 |
| **Level of education of individual** | | | | | | |
| No formal schooling | Ref | | | Ref | | |
| Less than 5 years of schooling | 1.48 | [1.15, 1.91] | 0.003 | 0.77 | [0.61, 0.98] | 0.034 |
| 6–10 years of schooling | 1.67 | [1.22, 2.28] | 0.002 | 0.64 | [0.48, 0.85] | 0.002 |
| More than 10 years of schooling | 1.41 | [0.94, 2.12] | 0.100 | 0.49 | [0.31, 0.78] | 0.002 |
| **Age group (in years)** | | | | | | |
| 60–69 | Ref | | | Ref | | |
| 70–79 | 0.79 | [0.64, 0.99] | 0.037 | 2.20 | [1.80, 2.68] | 0.000 |
| 80+ | 0.62 | [0.38, 1.02] | 0.061 | 2.61 | [1.81, 3.76] | 0.000 |
| **Working status of individual** | | | | | | |
| Not working | Ref | | | Ref | | |
| Working | 1.33 | [0.99, 1.79] | 0.061 | 0.64 | [0.50, 0.83] | 0.001 |
| **Individual received old age pension** | | | | | | |
| No | Ref | | | Ref | | |
| Yes | 1.29 | [0.93, 1.79] | 0.130 | 1.36 | [0.98, 1.90] | 0.070 |
| **Individual suffering from chronic diseases** | | | | | | |
| No | Ref | | | Ref | | |

*(Continued)*

**Table 3.** (Continued)

| Characteristics in wave-I | Change in headship status during wave-II | | | | | |
|---|---|---|---|---|---|---|
| | Gained headship | | | Lost headship | | |
| | OR | 95% CI | P-value | OR | 95% CI | P-value |
| Yes | 0.79 | [0.62, 1.01] | 0.062 | 1.21 | [0.96, 1.52] | 0.100 |
| **Gender of household head** | | | | | | |
| Male | Ref | | | Ref | | |
| Female | 1.20 | [0.71, 2.02] | 0.490 | 1.95 | [1.43, 2.66] | 0.000 |
| **Household wealth quintile** | | | | | | |
| Poorest | Ref | | | Ref | | |
| Poor | 1.02 | [0.76, 1.37] | 0.900 | 1.04 | [0.75, 1.43] | 0.830 |
| Medium | 0.86 | [0.63, 1.17] | 0.340 | 0.84 | [0.60, 1.17] | 0.290 |
| Rich | 0.80 | [0.58, 1.11] | 0.180 | 0.82 | [0.57, 1.16] | 0.260 |
| Richest | 0.66 | [0.46, 0.94] | 0.023 | 0.89 | [0.60, 1.32] | 0.560 |
| **Household BPL status** | | | | | | |
| Not poor | Ref | | | Ref | | |
| Poor | 0.97 | [0.77, 1.23] | 0.820 | 1.20 | [0.94, 1.52] | 0.140 |
| **Caste of household** | | | | | | |
| Scheduled Tribes | Ref | | | Ref | | |
| Scheduled Castes | 0.76 | [0.52, 1.12] | 0.170 | 1.15 | [0.73, 1.82] | 0.540 |
| Other Backward Classes | 0.58 | [0.40, 0.85] | 0.004 | 1.34 | [0.87, 2.06] | 0.190 |
| Others | 0.64 | [0.44, 0.95] | 0.025 | 1.17 | [0.74, 1.84] | 0.500 |
| **Religion of household** | | | | | | |
| Hindu | Ref | | | Ref | | |
| Muslim | 1.15 | [0.84, 1.56] | 0.390 | 0.87 | [0.63, 1.20] | 0.400 |
| Others | 0.91 | [0.66, 1.26] | 0.570 | 1.02 | [0.71, 1.47] | 0.900 |
| **Place of residence** | | | | | | |
| Rural | Ref | | | Ref | | |
| Urban | 1.39 | [1.10, 1.75] | 0.005 | 1.03 | [0.81, 1.31] | 0.810 |
| **Country regions** | | | | | | |
| Northern | Ref | | | Ref | | |
| North Eastern | 0.93 | [0.52, 1.65] | 0.810 | 0.81 | [0.40, 1.65] | 0.560 |
| Central | 0.88 | [0.64, 1.20] | 0.410 | 1.65 | [1.17, 2.33] | 0.004 |
| Eastern | 0.69 | [0.51, 0.92] | 0.012 | 1.53 | [1.14, 2.05] | 0.004 |
| Western | 0.88 | [0.67, 1.14] | 0.330 | 0.87 | [0.62, 1.22] | 0.420 |
| Southern | 0.90 | [0.70, 1.15] | 0.400 | 2.48 | [1.93, 3.19] | 0.000 |
| **Analytical sample size** | **5,334** | | | **5,193** | | |

Note–(a) OR: Odds ratio, 95% CI: 95% Confidence Interval; (b) (Ref) denotes reference category.

Further, married older adults in wave-I had 3.57 times [CI: 2.56,4.97] higher and 0.35 times [CI: 0.25,0.49] lower likelihood of gaining and losing their headship in wave-II respectively. With the increasing number of married adults in a household, the chances of losing headship among older adults also increases in wave-II. Older adults who had more than 10 years of schooling education in wave-I experience 0.49 [CI: 0.31,0.78] times lesser chance of losing headship in wave-II. Moreover, working older adults were 0.64 [CI: 0.50,0.83] times lesser likely to lose their headship in wave-II. Effect of greying was also noticed as older adults age 70–79 and 80+ years in wave-I had 2.20 [CI: 1.80,2.68] and 2.61 [CI: 1.81,3.76] times higher likelihood of losing their headship in wave-II respectively. Female headed household in wave-I

had 1.20 [CI: 0.71,2.02] and 1.95 [CI: 1.43,2.66] times higher chance of gaining and losing headship in wave-II respectively. Older adults coming from poor households also experienced higher headship loss in wave-II. Moreover, older adults of southern regions of India in wave-I had 2.48 [CI: 1.93,3.19] times higher likelihood of losing headship in wave-II, as compared to their counterparts from northern India.

## Discussion

Using a panel dataset of the India Human Development Survey, the present study found that within seven years, 6.8% of households had transitioned from extended to single generation and nearly 1.9% of households from nuclear to a single generation. The results were consistent with previous studies where it was found that the proportion of one-person households, which in our study comes under single generation households, was increasing due to increased urbanization and labor migration [31, 32]. Similarly, it was also revealed that single generation households had increased from 10.9% to 17% in almost seven years and nuclear family structure decreased from 12.1% to 6.7%, which is again an interesting finding when living arrangements of older adults are considered. The results were consistent with the findings of the Indian Demographic and Health Survey data which revealed that the nuclear family structure among older adults had significantly decreased from 1992–93 to 2015–16 [33].

Importantly, changes in the family demography plays a major role in the headship transition of older adults. Studies found that an increased quest for privacy among the young led to a sharp rise in the early marriages and that in turn, raised the proportion of the young male household heads [34]. In this regard, the marital statuses of the children which result in disaggregation of the household, contribute to the loss of headship among older parents and the automatic attribution of headship to the senior member of the family cases to be the norm [35]. The present analysis also shows that older adults who live in the nuclear family had higher odds of losing their household headship in comparison to older adults who live as single generation i.e., either alone or with a spouse. The results are consistent with previous studies which found that in India one-person household headship is increasing especially among older women [32]. Moreover, in some cases, the adult children become household heads but prefer a separate living arrangement and move away from their older parents. Determining the proportion of these cases in the Indian context is important and its impact on older adults' wellbeing is subject to further investigation [36, 37]. The loss of headship in nuclear and joint/extended family was high probably because as evidence suggests, when the son gets economically dependent or gets married, then the probability of loss of headship increases among older adults [9]. There is a huge scope to look into the household dynamics i.e., whether the son is married or not or whether he is financially dependent or not which would, in turn, affect the headship status of the older adult. Moreover, it was found that grown-up children tend to make decisions as they become the main breadwinners in the household, and in turn, the decision-making power of older persons in India has been declining [38]. This suggests that in the Indian scenario there is a voluntary transfer of headship to the economically active sons.

The older adults who were currently married had higher odds of gaining household headship in comparison to older adults who were not currently married. The relationship can be understood in the context that if in older age the spouse is dead, it may have adverse consequences on the surviving older adult. Widowhood may affect mental health and social participation and result in behavioral issues in the surviving partners [39, 40]. These changes are likely to affect the decision-making skills and hence the loss of headship is high among older adults who were widowed/separated/never married/divorced. However, as evidence suggests, in most cases, the older member of the family is deemed to the formal head and the actual

management of the house is run by the adult son [41]. Moreover, how the loss of a spouse affects the value of older adults in the later years of life is the prime area of investigation, because if the value deteriorates then it may lead to loss of headship and vice-versa.

Previous studies argued that older adults with higher educational status had an increased probability of losing their household headship [33]. The present results are however, inconsistent and suggest that older adults with higher educational attainment had lower odds of losing headship. Hence, the loss of headship among educated older adults is an area that warrants further investigation. Besides, the age gradient was considered as the most important predictor for loss of headship among older adults. As people grew old they are considered a liability by the younger family members, and hence had higher chances of losing headship [42].

Further, we observed that older adults who were working were less likely to lose their headship which supported the notion that older adults with better economic status had the greater bargaining power to choose their status in the family [43, 44]. In the present study, it was also found that older adults with a pension had higher chances to lose headship. The reason would be that the beneficiary's pension would not be sufficient to prove his position in the family with a secured economic status. In this context, although the person may not be the head but may make household decisions, the concept of nominal head and functional head may be an interesting factor for further investigation. However, the finding is contrary to, a study that found that older adults who receive pension had reported better social status, independence and better quality of life [45]. Also, previous studies have shown that poor older adults had higher chances of losing power and headship since they are considered as no more capable of contributing to the household economy and becoming a burden on other family members [43, 46, 47]. Nonetheless, our study found that older adults coming from a household below the poverty line (BPL) had higher odds of losing their headship.

Furthermore, older adults with chronic conditions had higher chances of losing household headship which is consistent with previous studies. The reason stated was that older adults with chronic diseases become dependent on their children or other relatives and hence lose their relevance in the family and household headship [9, 48]. The movement of older adults from one household to other due to worsening health conditions affects the headship status [9]. Also, the worsening condition would probably affect their decision-making skills; hence, loss of headship may happen at older ages.

Women were more likely to gain headship at older ages as they are more likely to be widowed and hence had higher chances to become head following the departure of their spouse [32]. Further, older adults from urban areas were having higher odds of gaining headship. The result was parallel with previous studies where it was argued that adults in their older ages from urban areas were more likely to be the household heads [9]. Older adults from the southern region of India had higher chances of losing the headship, which can be explained by the increasing number of nuclear families in southern states of India [49]. However, the reason behind is yet to be investigated further and a dearth of literature on the regional variations in the headship status of older populations provides a window for further investigation.

The study had certain limitations which should be mentioned. Firstly, the data is eight years old, therefore, one should be cautious while generalizing the results at present; however, the association may still persist. Secondly, the status of the son is not available in the dataset. This could have been one of the important predictor of gain or loss of headship status for older adults as if the son is the sole breadwinner of the family, he would have been the head of the household or the main decision maker of the household. Lastly, chronic diseases were self-reported and may add to the response bias. Moreover, future studies examining the association between family structure and transition in headship status can look into the interaction effects of explanatory variables in the observed associations. However, acknowledging the strengths,

the study provides a detailed information on an utmost important concept of gain and loss of headship among older adults in India. Although the data is older, IHDS (2004–05 and 2011–12) was the only publicly available panel dataset that would allow us to examine the headship transition among older adults in India. This also indicates the need for a new comprehensive quantitative dataset that covers multiple aspects of family demography in India.

## Conclusion

It can be concluded that the loss of household headship was significantly associated with the type of family structure. The present study re-establishes the importance of individual and household level determinants of headship status, such as income, health, and the fact that privacy and independence in later ages are being increasingly achieved at the cost of weakening the family ties. In this regard, the emergence of an increase in single generation households along with declining joint families has changed the character of household headship for older individuals. However, living in a single-generation household may preferably be encouraged among older adults than their living in a complex household without headship. Thus, public intervention may support independent living within the older population through housing policies that create additional choices presented to older adults making residential decisions.

Nonetheless, the research questions remain such as whether the cultural differences, availability of children, and objective health indicators have affected the household headship status late in life than the indicators the analysis of which yielded the current conclusions. Moreover, a potential line of inquiry would be to distinguish the determining factors and changes in household headship rate in the people from younger age groups and its effect on the headship status of their older parents. Studies related to India's aged population from the family perspective are becoming extremely important, given the increasing proportion of aged persons in the population composition of the nation.

## Author Contributions

**Conceptualization:** Shobhit Srivastava.

**Data curation:** Ronak Paul.

**Formal analysis:** Ronak Paul.

**Methodology:** Ronak Paul.

**Supervision:** Shobhit Srivastava.

**Validation:** Rashmi Rashmi.

**Writing – original draft:** Muhammad Thalil, Rashmi Rashmi, Ronak Paul.

**Writing – review & editing:** Shobhit Srivastava, Muhammad Thalil, Rashmi Rashmi.

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
