## [Decision Letter · Decision Letter 0]

1 Apr 2021

PONE-D-20-38024

Association of Family Structure with Gain and Loss of Household Headship among Older Adults of India: Analysis of Panel Data

PLOS ONE

Dear Dr. Paul,

Thank you for submitting your manuscript to PLOS ONE. After careful consideration, we feel that it has merit but does not fully meet PLOS ONE’s publication criteria as it currently stands. Therefore, we invite you to submit a revised version of the manuscript that addresses the points raised during the review process.

ACADEMIC EDITOR: Revise and Resubmit.

We look forward to receiving your revised manuscript.

Kind regards,

Srinivas Goli, Ph.D.

Academic Editor

PLOS ONE

Journal Requirements:

Additional Editor Comments (if provided):

Considering the reviewers opinion, I am going with a decision of major revision. I have following suggestions. Please include a variable in the background characteristics that number of married adults in the family and the number of unmarried adults and children in the family. Bring some sociological or family demography literature on why and when headship transition in the households takes place and whether your analyses are supporting or rejecting those literature and hypothesis? Also, can you cut-down the length of headings in the manuscript and Tables.

Reviewers' comments:

Reviewer's Responses to Questions

**Comments to the Author**

1. Is the manuscript technically sound, and do the data support the conclusions?

Reviewer #1: Partly

Reviewer #2: Partly

2. Has the statistical analysis been performed appropriately and rigorously? 

Reviewer #1: Yes

Reviewer #2: Yes

3. Have the authors made all data underlying the findings in their manuscript fully available?

Reviewer #1: Yes

Reviewer #2: Yes

4. Is the manuscript presented in an intelligible fashion and written in standard English?

Reviewer #1: Yes

Reviewer #2: Yes

5. Review Comments to the Author

Reviewer #1: This article addresses the important topic the association of family structure with gain and loss of household headship among older adults of India from the panel data of IHDS wave-I and wave-II. The outcome indicators for the study was the gain in headship and loss in headship with other explanatory and control variables. Authors have concluded that the loss of household headship was significantly associated with the changes in the family structure. The results of the multivariate analysis found that older adults from economically poor background and older adults with chronic disease conditions had higher chances of losing headship. Although, the paper has some limitations. Here some broad suggestions:

1. Page no. 8 (208-209) Authors have mentioned that they have found the interaction effect of marital status, income, education level and household BPL status. But, did not find any explanation behind it. Authors would have added table for the changes in the characteristics of older adults in both rounds of data sets which are directly affecting the household headship status like marital status, working status and Household BPL status. The interaction effect of these variables can be looked up. The work status and marital status decides the living arrangement of older adults, and living arrangement affects the household headship status. If any older adults who was not in poor in wave-I but entered in poor (BPL) category or vice-versa, this can affect the work status of the older adults.

2. Page no. 7 (164-167). The categorization of marital status variable is not clear. It can be written in more clear way.

3. Page no. 17-18. In the discussion section, culture of the Indian society can be discussed more in term of voluntary transfer of headship to their economically active sons.

Minor comment

1. Page no. 16. ® Symbol can be assigned against the reference category in the table 3 as mentioned in note.

Reviewer #2: The authors conclude that the loss or gain of headship was associated with changes in the family structure but their analysis does not reflect this. The multivariate results primarily show the odds of gaining or losing headship status given that one belongs to a particular family type but do not analyze whether the gain or loss can be attributed to a change in family type. For instance, the older person might have lost headship due to a change from nuclear to joint/extended family.

It is unclear the reason for using characteristics in wave I when these might have changed in wave II. Maybe it is an issue with how it has been conceptualized. Can authors explain this?

For the multivariate analysis, authors could consider running two models for each of the outcomes. The first model could contain only the primary factors - family type - since it is the main association authors wish to test. The second model will then account for the effect of control variables. The odds ratio and CI can be reported in one column (in order to get enough space for the two models). Also, authors could indicate in the table the reference category for the independent variables as Ref or 1.00. Using ® (which did not appear at all in table 3) is somehow non-academic. In table 3, square brackets should be used instead when reporting CIs. Also, authors should use a comma instead of "dash". eg. [0.67, 1.40]

Under the statistical methods, can authors explain their reason for testing interactions? in my opinion, there may not be a need to write on interactions. This is because the objective of this study was not to investigate the interactions between specific variables.

line 227 - .. wave I increased instead of "were increased".

line 236 - authors could consider using "no formal education or no formal schooling" instead of no schooling education

Authors could also briefly explain how the limitations link to their study. For instance, how is the non inclusion of the dependency status of the son a limitation for this study.

6. PLOS authors have the option to publish the peer review history of their article (what does this mean?). If published, this will include your full peer review and any attached files.

Reviewer #1: No

Reviewer #2: No

---

## [Author Response · Author response to Decision Letter 0]

8 Apr 2021

Reviewer’s Comment

Reviewer #1: This article addresses the important topic the association of family structure with gain and loss of household headship among older adults of India from the panel data of IHDS wave-I and wave-II. The outcome indicators for the study was the gain in headship and loss in headship with other explanatory and control variables. Authors have concluded that the loss of household headship was significantly associated with the changes in the family structure. The results of the multivariate analysis found that older adults from economically poor background and older adults with chronic disease conditions had higher chances of losing headship. Although, the paper has some limitations. Here some broad suggestions:

1. Page no. 8 (208-209) Authors have mentioned that they have found the interaction effect of marital status, income, education level and household BPL status. But, did not find any explanation behind it. Authors would have added table for the changes in the characteristics of older adults in both rounds of data sets which are directly affecting the household headship status like marital status, working status and Household BPL status. The interaction effect of these variables can be looked up. The work status and marital status decides the living arrangement of older adults, and living arrangement affects the household headship status. If any older adults who was not in poor in wave-I but entered in poor (BPL) category or vice-versa, this can affect the work status of the older adults.

Response: Dear reviewer, thank you very much for pointing this out. However, keeping in mind the comment given by the editor and reviewer #2 we have refrained from examining interaction effects further. However, we have included this point as a scope for further research in the “Discussion” section (Page 20, line 380-382).

2. Page no. 7 (164-167). The categorization of marital status variable is not clear. It can be written in more clear way.

Response: Thank you for the suggestion. Accordingly, we have modified the line to (Page 7, line 167-169): 

“Individuals who were “single”, “widowed”, “separated/divorced”, “spouse absent” and “no gauna” were recoded into the “Not currently married” category; otherwise, they were included in the “Currently married” category.”

3. Page no. 17-18. In the discussion section, culture of the Indian society can be discussed more in term of voluntary transfer of headship to their economically active sons.

Response: comment incorporated (Line no. 323-327)

Minor comment

1. Page no. 16. ® Symbol can be assigned against the reference category in the table 3 as mentioned in note.

Response: Thank you very much for pointing this out. We have denoted the reference category in Table 3 by assigning the symbol (Ref).

Reviewer #2: The authors conclude that the loss or gain of headship was associated with changes in the family structure but their analysis does not reflect this. The multivariate results primarily show the odds of gaining or losing headship status given that one belongs to a particular family type but do not analyze whether the gain or loss can be attributed to a change in family type. For instance, the older person might have lost headship due to a change from nuclear to joint/extended family.

Response: Dear sir, Thank you for the very important insight. I completely agree with your comment. I have rephrased the first line of conclusion. (Line no. 388-389)

1. It is unclear the reason for using characteristics in wave I when these might have changed in wave II. Maybe it is an issue with how it has been conceptualized. Can authors explain this?

Response: We agree that characteristics such as marital status, household wealth quintile, and household BPL status might change between round-I and round-II. However, our objective is to check whether the characteristics in round-I were associated with the change in household headship status between round-II. Accordingly, the study has been conceptualized. In the present study we just tried to find the predictors of loss or gain of headship among older adults. 

2. For the multivariate analysis, authors could consider running two models for each of the outcomes. The first model could contain only the primary factors - family type - since it is the main association authors wish to test. The second model will then account for the effect of control variables. The odds ratio and CI can be reported in one column (in order to get enough space for the two models). Also, authors could indicate in the table the reference category for the independent variables as Ref or 1.00. Using ® (which did not appear at all in table 3) is somehow non-academic. In table 3, square brackets should be used instead when reporting CIs. Also, authors should use a comma instead of "dash". e.g. [0.67, 1.40]

Response: We agree with the reviewer’s suggestion. Therefore, keeping in mind the editor’s and the reviewer’s suggestion regarding formatting of Table 3 we have denoted CI within square brackets separated by a comma in between. Further, reference category for independent variables has been shown using Ref. 

As per your suggestion, we considered running two separate models with the first model showing the bivariate association of family type in round-I with headship change between round-I and round-II and the second model accounting for the effect of control variables. We humbly point out that bivariate association has of family type in round-I with headship change between round-I and round-II has already been examined in Table 2. Evidently, we did not repeat the same in Table 3.

3. Under the statistical methods, can authors explain their reason for testing interactions? in my opinion, there may not be a need to write on interactions. This is because the objective of this study was not to investigate the interactions between specific variables.

Response: Dear reviewer, thank you for pointing this out. Accordingly, we have omitted the section for testing of interactions in the revised manuscript. 

4. line 227 - ... wave I increased instead of "were increased".

Response: Thank you for pointing this out. We have corrected the sentence to (Page 9, line 223-224):

As shown in figure 2, the percentage of older adults who were residing in a single generation household in wave-I increased to nearly 6% in wave-II.

5. line 236 - authors could consider using "no formal education or no formal schooling" instead of no schooling education.

Response: Thank you for pointing this out. We have corrected the sentence to (Page 10, line 232): 

Further, 67% of older adults were married, 60% had no formal education and 85% were unemployed during wave-I.

6. Authors could also briefly explain how the limitations link to their study. For instance, how is the non-inclusion of the dependency status of the son a limitation for this study.

Response: Comment incorporated. (Line 375-386)

---

## [Editor Report · Decision Letter 1]

11 Apr 2021

PONE-D-20-38024R1

Association of Family Structure with Gain and Loss of Household Headship among Older Adults in India: Analysis of Panel Data

PLOS ONE

Dear Dr. Paul,

Thank you for submitting your manuscript to PLOS ONE. After careful consideration, we feel that it has merit but does not fully meet PLOS ONE’s publication criteria as it currently stands. Therefore, we invite you to submit a revised version of the manuscript that addresses the points raised during the review process.

ACADEMIC EDITOR: Dear Authors, I have asked you to include some variables in the background characteristics: ‘the number of married, unmarried adults and children in the family. Without understanding the age and marital status composition of other household members, it is difficult to explain why and when headship transition takes place in families. In general, the marriage of children is the first break-point for older adult headship. What kind of headship loss you’re talking about for single-person older households, whom he/she is heading. What do you understand by headship? I think you need to put some logic while planning the analyses.  Also, this paper is still silent on the question that what had happened between the waves that an older adult has to give up his/her headship? IHDS allows adding such variables. Also, bring some sociological or family demography literature on why and when headship transition in the households takes place and whether your analyses are supporting or rejecting that hypotheses? Also, can you cut down the length of headings in the manuscript and Tables? If you feel these suggestions are not relevant to your paper, please respond to them.

We look forward to receiving your revised manuscript.

Kind regards,

Srinivas Goli, Ph.D.

Academic Editor

PLOS ONE

Journal Requirements:

Additional Editor Comments (if provided):

Dear Authors, I have asked you to include some variables in the background characteristics: ‘the number of married, unmarried adults and children in the family. Without understanding the age and marital status composition of other household members, it is difficult to explain why and when headship transition takes place in families. In general, the marriage of children is the first break-point for older adult headship. What kind of headship loss you’re talking about for single-person older households, whom he/she is heading. What do you understand by headship? I think you need to put some logic while planning the analyses. Also, this paper is still silent on the question that what had happened between the waves that an older adult has to give up his/her headship? Due to death of a male partner, by default female become head in a two member family, is consider as gaining headship? IHDS allows adding such variables. Also, bring some sociological or family demography literature on why and when headship transition in the households takes place and whether your analyses are supporting or rejecting that hypotheses? Also, can you cut down the length of headings in the manuscript and Tables? If you feel these suggestions are not relevant to your paper, please respond to them.

---

## [Author Response · Author response to Decision Letter 1]

15 Apr 2021

MANUSCRIPT TITLE: Association of Family Structure with Gain and Loss of Household Headship among Older Adults in India: Analysis of Panel Data

SUBJECT: Response to reviewers

Respected editor, 

Thank you for giving us the opportunity of submitting an improved version of our manuscript for publication in the PLOS One. We are highly grateful to receive your insightful comments and suggestions. We appreciate the time and effort that you have put forward to provide valuable feedback that has significantly improved our paper. Kindly note that we have incorporated the changes that were suggested. The modifications have been shown using track changes within the revised manuscript. Please see below for a point-by-point response to each of the comments and suggestions.

Hoping that you and your family members are safe and sound in these challenging times,

Yours Sincerely,

Authors

 

Editor’s Comments:

1. Dear Authors, I have asked you to include some variables in the background characteristics: ‘the number of married, unmarried adults and children in the family. Without understanding the age and marital status composition of other household members, it is difficult to explain why and when headship transition takes place in families. In general, the marriage of children is the first break-point for older adult headship.

Response: Dear editor thank you for the suggestion. We have included variables showing the number of married adults, number of unmarried adults and number of children in the family, in our analysis. Accordingly, the “Data and methods”, “Results” and “Discussion” section and the tables have been updated. 

2. What kind of headship loss you’re talking about for single-person older households, whom he/she is heading. What do you understand by headship? I think you need to put some logic while planning the analyses.

Response: Dear sir, we are not considering single-person older households. For the headship to be loosed, at least 2 persons should reside in a single household. Secondly, headship means the person who is heading the household, taking all the major decisions. In the large-scale survey, we generally ask all the household information from the household head only. Therefore, the dynamics of headship in IHDS-1 and IHDS-2 was observed from the present analysis. 

3. Also, this paper is still silent on the question that what had happened between the waves that an older adult has to give up his/her headship? IHDS allows adding such variables.

Response: Dear editor, thank you for pointing this out. Kindly note that in this paper we aim to analyse the association of family structure and other explanatory variables in wave-I with their change in headship status between wave-I and wave-II. The same has also been pointed out by Reviewer #2. We do not try to answer the question that what had happened between the waves that an older adult has to give up his/her headship.

We know that using the “IHDS tracking sheet” we can determine whether an older adult died or was untraceable between round-I and round-II. Since we are using the panel data, it effectively means that we are analysing only those adults who were alive and traceable by IHDS staff after wave-I and during wave-II. Further, we have already noted the following in our manuscript (Page 5, line 114-117):

During IHDS wave-I, there were 17,906 individuals, aged 60 years and above, whom we refer to as older adults in this study. Among them, 4,736 older adults were not alive and 2,643 older adults were lost to follow-up during wave-II. Thus, our current study is based on panel data of 10,527 older adults.

4. Also, bring some sociological or family demography literature on why and when headship transition in the households takes place and whether your analyses are supporting or rejecting that hypotheses?

Response: Comment incorporated. 

5. Also, can you cut down the length of headings in the manuscript and Tables?

Response: Thank you. We have shortened the length of headings in the manuscript and tables. 

6. If you feel these suggestions are not relevant to your paper, please respond to them.

Response: Dear editor, on a contrary note we believe that all the suggestions given by you have improved our manuscript by leaps and bounds. Therefore, we have carefully studied, incorporated and then responded to each of them.

---

## [Editor Report · Decision Letter 2]

21 May 2021

Association of Family Structure with Gain and Loss of Household Headship among Older Adults in India: Analysis of Panel Data

PONE-D-20-38024R2

Dear Dr. Paul,

We’re pleased to inform you that your manuscript has been judged scientifically suitable for publication and will be formally accepted for publication once it meets all outstanding technical requirements.

Kind regards,

Srinivas Goli, Ph.D.

Academic Editor

PLOS ONE

Additional Editor Comments (optional):

Revisions are acceptable but authors have to send a clean version after incorporating two final suggestions: 1. Remove track changes and read the paper carefully for language, grammar, typos and syntax. 2. I am giving additional reference of a paper that recently published. Can you contextualise your paper topic in a broader family demographic literature. Add two or three lines in limitations what kind of data is needed in future to study demographic issues in family perspective. This helps your to live beyond the life span of IHDS round 1 and 2.

Chakravorty S, Goli S, James KS. Family Demography in India: Emerging Patterns and Its Challenges. SAGE Open. 2021 Apr;11(2):21582440211008178.
---

## [Editor Report · Acceptance letter]

27 May 2021

PONE-D-20-38024R2 

Association of Family Structure with Gain and Loss of Household Headship among Older Adults in India: Analysis of Panel Data 

Dear Dr. Paul:

I'm pleased to inform you that your manuscript has been deemed suitable for publication in PLOS ONE. Congratulations! Your manuscript is now with our production department. 

Kind regards, 

on behalf of

Dr. Srinivas Goli 

Academic Editor

PLOS ONE